# Robotic-Assisted Microsurgery and Its Future in Plastic Surgery

**DOI:** 10.3390/jcm11123378

**Published:** 2022-06-13

**Authors:** Matthias M. Aitzetmüller, Marie-Luise Klietz, Alexander F. Dermietzel, Tobias Hirsch, Maximilian Kückelhaus

**Affiliations:** 1Department of Trauma, Hand and Reconstructive Surgery, Division of Plastic Surgery, University Hospital Muenster, 48149 Muenster, Germany; aitzetmueller.m@hotmail.com (M.M.A.); marie.luise.klietz@gmx.de (M.-L.K.); alex.dermietzel@gmx.de (A.F.D.); tobias.hirsch@uni-muenster.de (T.H.); 2Department of Plastic, Reconstructive and Aesthetic Surgery, Hand Surgery, Fachklinik Hornheide, 48157 Muenster, Germany; 3Division of Plastic Surgery, Institute of Musculoskeletal Medicine, University Hospital Muenster, 48149 Muenster, Germany

**Keywords:** robotic-assisted surgery, microsurgery, robotic microscope

## Abstract

Within the last 20 years, robotic-assisted surgeries have been implemented as routine procedures in many surgical fields, except in plastic surgery. Although several case series report promising results, technical and economic aspects have prevented its translation into clinical routine. This review is based on a PubMed and Google Scholar database search, including case reports, case series, clinical and preclinical trials, as well as patents. Past, recent approaches, ongoing patents, as well as eight specific systems for robotic-assisted microsurgery and their potential to be translated into a clinical routine, are described. They may lay the ground for a novel field within plastic surgery. This review provides an overview of the emerging technologies and clinical and preclinical studies and discusses the potential of robotic assistance in the field of plastic surgery.

## 1. Introduction

All surgeries aim to simultaneously decrease invasiveness and enhance effectiveness. There has been a great deal of enthusiasm and effort in many surgical fields to introduce robotics in support of these aims. However, the generally very innovative field of plastic surgery has not integrated robotics into its routine clinical work yet.

The invention of laparoscopy in 1985 represents a big leap in decreasing invasiveness [1,2] (Figure 1).

The advances in laparoscopic procedures in the 1980s were at first received with criticism but represented a big leap in decreasing the invasiveness of surgeries.

Clinical benefits include smaller incisions, thereby reducing trauma, the risk for surgical site infections, and hospitalization time. Additionally, laparoscopic surgeries lead to enhanced patient satisfaction, including a better cosmetic outcome, decreased postoperative pain, as well as a shorter absence from work. Starting with the first cholecystectomy, laparoscopic surgeries have become an integral part of specialties such as urology, abdominal surgery, gynecology, and cardiac surgery [3,4].

Ten years later, another surgical milestone was reached with the performance of the first robotic-assisted surgery (RAS). The first RAS represented a cholecystectomy performed in 1997 [5], followed by a mitral valve replacement 1 year after [6]. RAS can be seen as an evolution of laparoscopic surgery. In addition to the minimally invasive access to the surgical field, they allow performing surgeries with a three-dimensional view, downscaling the surgeon’s movements, tremor elimination, and additional axes of movement.

The first transatlantic telesurgery, the Lindberg operation, was carried out in 2001. The surgeon was located in New York City, the USA, operating on a patient in Strasbourg, France. This can be considered the next immense milestone in RAS. The novel field of telemedicine and telesurgery opened treatment modalities and surgical expertise to remote areas.

However, not only has availability changed but also surgical systems. While initially robots were seen as “unintelligent machines” that are “rooted by the strengths and weaknesses of their predecessors” [7], this changed a lot. Due to radical changes and recent advances in the field of artificial intelligence, modern robots are potentially able to distinguish between different tissues. Artificial neural networks can learn and optimize specific operations. When looking at recent developments within other medical fields, artificial intelligence has shown to be partially more effective than conventional human-based medicine. As an example, ophthalmological screening for diabetic retinopathy by an automatic camera system has shown higher sensitivity than screening performed by ophthalmologists [8,9].

Following these trends, the combination of robotic-assisted surgery in combination with artificial intelligence has the potential to change the position of the surgeon from an active operator toward a supervisor within the upcoming years.

## 2. Materials and Methods

This review is based on a database search, including case reports, case series, clinical and preclinical trials, as well as patents. Data collection was performed by using the databases “Embase, MEDLINE (via PubMed), Web of Science, and Google Scholar”. Search terms included “Robotic assisted microsurgery”, “Robotic assisted surgery”, “Symani”, “robotic microscope”, “MUSA”, and “Robotic assisted surgery”.

## 3. Results

### 3.1. Surgical Systems

The field of RAS is a novel field underlying rapid change. Most of these developments have originally been invented for aerospace and military purposes. Nevertheless, only two surgical robots have made their way into clinical routine up to now. The first system that was FDA approved in 2000 was the da Vinci surgical system (Intuitive Surgical, Sunnyvale, CA, USA). One year later, the Zeus robotic system (Computer Motion Inc., Goleta, CA, USA) received FDA approval and merged into the market. Differences included the number of arms available (3–4 within the Da Vinci and 3 of the Zeus) and a voice-activated camera system. While the first cholecystectomies were performed using the da Vinci, the initial transatlantic surgery was performed using the Zeus system.

Being rivals for decades, these systems were combined when Computer Motion Inc. and Intuitive Surgical merged in 2003.

Through further development, the potential of RAS was recognized by other specialties. In 2007, the first microsurgical anastomosis was performed by the da Vinci [10]. The anastomosis, performed for the first time with the help of a robot, was part of an autologous breast reconstruction using a TRAM flap. While the venous anastomosis was performed manually, arterial anastomosis of the deep inferior epigastric artery to the internal mammary artery was performed by the surgeon using the da Vinci. Since then, harvesting of free flaps such as the DIEP has been performed multiple times using the da Vinci [11,12,13,14].

This can be considered the beginning of RAS in microsurgery. Already this initial approach showed the limitations of existing systems, given by the insufficient optical magnification and instrument size. Therefore, robotic platforms customized for the needs of microsurgery were created. These systems have multifaceted goals: (a) to effectively filter tremors, (b) to downscale movements beyond the da Vinci’s capacities, (c) to provide sufficient magnification, and (d) to be easy to integrate into the OR settings for microsurgical procedures. The Microsurgical Robot (MSR), the MUSA (MicroSure, Eindhoven, Netherlands—Figure 2), and the Symani System (MMI, Pisa, Italy—Figure 3) were developed to meet the aforementioned criteria. These systems are currently in a phase of clinical evaluation and potentially impact the field of microsurgery.

### 3.2. Robotic-Assisted Plastic Surgery

The use of robots within the field of plastic surgery has been fascinating since the early beginnings of RAS. In 2005, the first free flap was harvested and anastomosed in a Minipig model by using the da Vinci system. Initial clinical case reports were published in 2007 when van der Hulst used the da Vinci system for microsurgical anastomosis [10]. Although they described the robotic-assisted anastomosis as taking longer than a normal anastomosis, they acknowledged a fast learning curve.

In 2014, Clemens et al. published the first cohort study about RAS in breast reconstruction [15]. They analyzed surgical outcomes of patients who underwent delayed immediate breast reconstruction after radiation therapy. Breast reconstruction was performed by using a pedicled latissimus dorsi flap either with open surgery or robotic-assisted minimally invasive surgery. They showed a lower complication rate (16.7% in RAS vs. 37.5% in total open) by eliminating the need for donor site incisions. However, the study population included 146 patients in total, with only 17 undergoing RAS, and was therefore too small to detect significant differences.

In 2017, the da Vinci system was used for a mastectomy and implant-based immediate breast reconstruction [16]. Therefore, Toesca et al. used a 2.5 cm axillary approach. For adequate preparation, an insufflator was connected, and then the mastectomy was performed. Again, and similar to previous studies, the authors reported a radical learning curve. They mentioned that the first approach took three times longer than the third approach. The surgical time of the final approach was comparable to a mastectomy without robotic assistance. Additionally, RAS allowed a reduction in side effects, such as bleeding, increased vascularization, and a simultaneously enhanced aesthetic outcome.

Since this year, many case reports or case series that prove the feasibility and safety of RAS in breast surgery have been published [17,18,19]. A recent monocohort study with 74 patients reported a complication rate of 4.3% [20], highlighting the enormous potential of robotics in mastectomy.

Over the last years, novel approaches have included the usage of the da Vinci system for flap harvesting. Bishop et al. [21] described the outcome of 21 patients 12 months postoperatively after harvesting DIEP vessels by using a small posterior incision as a safe and reliable technique. Other studies have described the benefits of a minimal fascial incision for flap harvesting when compared with the traditional approach [11,12,13,14].

The da Vinci system has been further used for transoral approaches, such as tumor resection and reconstruction by local or free flaps. Therefore, the robotic system facilitated the anastomosis by easier access and allowing surgeries in limited space. Several case series and reports show this “TORS” (transoral robotic surgery) to be safe [22,23,24,25,26].

### 3.3. Robotic-Assisted Microsurgery, Supermicrosurgery, and Nanomicrosurgery

There is a large number of novel robotic platforms that are being developed for minimally invasive applications, such as MicroSurge, ViaCath, SurgiBottm, SPORTtm, MASTER, and SPIDER. However, there are only a few systems that are mainly focused on providing better performance in microsurgery. As microsurgery is being further developed toward supermicrosurgery, the physical limits for human operators without robotic support are becoming more evident and cannot be compensated by enhanced magnification. Complete tremor elimination and motion scaling are becoming more important while the focus shifts away from the need for minimally invasive access to the surgical site. In particular, lymphovenous anastomosis and super-thin free flaps, both of which are being performed more and more frequently, require the highest precision achievable.

The robotic system MicroSure MUSA is capable of using standard micro-/supermicrosurgery instruments without the need for expensive customized tools. This leads to lower costs and easier integration. The entire robotic system is attached to the OR table and can easily be brought into and removed from the surgical site. This system enables flexibility among the different elements of a microsurgery procedure, where, typically prior to and after microsurgery, a larger field of action is required. Recently, van Mulken et al. reported the first-in-human use of the MUSA system for the treatment of breast cancer-related lymphedema in a randomized pilot trial [27]. Although operation times were longer and anastomosis scored lower using the Structures Assessment of Microsurgery Skills (SAMS) and University of Western Ontario Microsurgical Skills Acquisition Instrument (UWOMSA), all anastomosis demonstrated patency in both groups. The 1-year follow-up study confirmed the feasibility of this approach with an anastomosis patency rate of 66.6% in the RAS group compared with 81.8% in the control group, indicating no significant differences [28]. Pricing for the MUSA system will be substantially lower than competitors due to the use of micro-/supermicrosurgery instruments.

The MMI’s Symani^®^ System with NanoWrist^®^ (MMI, Pisa, Italy) instrumentation followed in 2020 [29]. It was designed for open microsurgery as well. This system uses miniaturized wristed single-use instruments that provide a range of motion comparable to the da Vinci system. The Symani System has motion scaling up to 20x and is based on a mobile cart, which can easily be moved to and away from the operating field. In contrast to the MUSA, it uses telemetric system control. Thus far, this system has been used in five clinical cases for the pharynx and lower leg reconstruction (personal communication with MMI, cases presented at Barcelona Breast Meeting 2020). Additionally, Lindenblatt et al. confirmed the usage of the Symani System in the field of plastic and reconstructive surgery by presenting successful Symani-assisted lympho–venous anastomosis and lymph node transfer [30].

Both systems have recently received the CE marking, enabling regular clinical application in Europe. Launch in the United States is planned for the future.

The costs for a Symani System are currently in the high six-figure range (as of May 2022), plus around €2000 for disposable instruments per procedure (reference: Personal communication with MMI).

### 3.4. Robotic Microscopes and Augmented Reality

The microsurgeon usually uses loupes and microscopes in order to achieve sufficient magnification for handling vessels or nerves. Usually, perforator identification and pedicle preparation are performed using loupes. For these specific tasks, the flexibility to choose different angles during preparation is of great importance. For vessel anastomosis, a mostly stationary procedure in which magnification is more important, most surgeons prefer using a microscope. Robotic microscopes provide the technology to combine the advantages of both. The RoboticScope by BHS Technologies (Figure 4) utilizes a high-definition camera system that is connected to an augmented reality headset, and a clear image with high magnification is projected in front of the surgeon’s eyes.

At the same time, motion tracking translates the surgeon’s head movements onto the camera system via a multi-axis robotic arm. In this way, great magnification and flexibility become joint forces. The surgeon controls magnification levels and other settings via head movements. The headset removes itself from the surgeon’s field of vision by head command as well, not requiring any manual manipulation and enabling a fast switch between macroscopic and microscopic preparation. The RoboticScope is the first system to achieve the CE marking for clinical application in Europe in 2020. Shortly after, our group successfully performed the world’s first microsurgical breast reconstructions employing deep inferior epigastric perforator (DIEP) and profunda artery perforator (PAP) flaps using this system. Additionally, other groups have successfully shown clinical usage by successfully performing an anastomosis [31], resection of intracranial tumors [32], or a tympanoplasty [33]. Acquisition costs for the system currently start at around €200,000 (as of May 2022) (reference: personal communication with BHS).

### 3.5. Benefits of RAS in Plastic Surgery

The benefits of RAS in plastic surgery are obvious. Conventional robotic systems allow 100% of tremor elimination. Additionally, motion scaling of 20:1 is possible, leading to higher precision, especially in microsurgery and supermicrosurgery.

The laparoscopic approach shows minimal invasiveness and significant scar reduction. Conventional surgeries such as harvesting of latissimus dorsi flap have been associated with high morbidity of the donor region and hyperdimensional scars. With robotic assistance, these surgeries can become almost invisible.

Additionally, as already shown in several TORS approaches, robotic assistance can facilitate surgeries, especially in areas that are difficult to access. Although all of these approaches have been performed with systems that have not specifically been designed for microsurgery, usage was safe and beneficial.

### 3.6. Negative Aspects of RAS in Plastic Surgery

Due to the enormous economic costs, the need for complex infrastructure, a specifically schooled staff, and most likely an interdisciplinary approach, currently available RAS systems are rarely available outside of specialized centers. Therefore, access to these systems is limited.

Several studies describe a relatively long learning curve of more than 4 years [34]. To overcome this limitation, standardized training and assessment such as the “Structured Assessment of Robotic Microsurgical Skills (SARMS)” should be implemented in clinical settings [35,36]. This has been validated and showed a learning curve in all participants. While RAS is broadly implemented in other disciplines, such as gynecology, general surgery, or ENT surgery, plastic surgeons are not using RAS on a regular basis. Therefore, implementation of RAS would lead to the initial prolongation of most surgeries.

Additional costs most likely cannot be compensated, potentially representing the same struggle for cost-effective treatment as with the DaVinci system.

## 4. Discussion

Despite a broad introduction of robotic applications in several surgical specialties, no robot-assisted procedures are routinely performed in the field of plastic surgery. Thus far, minimally invasive vessel preparation is the main application performed in a few centers around the globe. As potential applications in reconstructive surgery where overcoming human physiological boundaries may facilitate better treatment options are increasing, the industry is shifting its focus toward customization for supermicrosurgery.

Recently, dedicated robotic systems for microsurgical applications have received CE certification to be implemented in the European market. These systems hold great potential to support procedures performed mainly by plastic surgeons. After the initial clinical experience, they will aim for FDA approval as well.

The surgical systems for the sole purpose of open microsurgery are designed around different philosophies. While some aim for a more cost-effective application than currently available robotic systems, others have focused on highly sophisticated miniaturized tools for one-time use. It will be interesting to observe whether all systems prevail or whether there will be mergers, such as Intuitive and Computer Motion with the DaVinci and Zeus systems.

In view of the extremely complex and cost-intensive training of competent microsurgeons, these systems potentially offer the possibility of benefiting from the experience of older surgeons for a longer period of time, even in the case of declining physical fitness. This circumstance may also contribute to the cost-effectiveness of microsurgery in the future. Simplification of supermicrosurgical procedures may allow more surgeons to perform the challenging operation and make treatments available for more patients.

In most countries, the high costs of such systems, for whose use no remuneration is envisaged for the time being, cannot be cross-financed by funds from regular operations. Costs are, therefore, one of the greatest obstacles to clinical use, and cost reduction needs to be considered essential for market establishment. In order to still play an active role in the further development of these systems, funding new technologies through the conduct of preclinical and clinical studies is a possibility. The performance of such studies is also essential to demonstrate a possible benefit or advantage as early as possible and to ensure appropriate remuneration for the use of the new technologies in the long term. Therefore, well-funded countries will most probably take the lead in advancing robotic microsurgery until improvements in manufacturing and more competition may result in a significant price decrease. If this process is successful, low-income countries may be able to also implement these technologies.

Special attention must be paid to good indications for the use of novel robotic systems. These may be lymphovenous anastomoses, followed by anastomoses in the context of free flap surgery, as well as replantation and nerve coaptation. For initial clinical application, the comparatively simple anastomoses of elective free flaps may be the entry point. Early identification of patient subgroups also needs to be emphasized since the new technologies could be of particular benefit here. As an example, the group of obese patients shall be mentioned, in whom difficult access to the operating site could be facilitated by a new robotic assist. The experience of other surgical specialties on the use of the DaVinci robot confirms here the need to collect data as early as possible in the context of studies and to achieve reimbursement, as well as the identification of patient subgroups that are most likely to benefit from the use of these new technologies.

The combination of visual and surgical master–slave systems may hold great potential for further improvement in flexibility. Moreover, the integration of more robotic arms may add to the efficiency of procedures and ultimately spare personnel, leading to a more cost-effective, one-surgeon approach. The optimal integration of an assistant remains to be demonstrated both by the surgical and the visual systems. Here, a learning curve leading to a second system generation can be expected.

In the future, it may be possible to establish a combined system allowing the plastic surgeon to perform specialized microsurgical procedures remotely from the office, providing services not tied to a single location.

## 5. Conclusions

Although RAS already represents a gold standard for many other surgical fields, its use in plastic surgery is still underestimated. While conventional systems such as the da Vinci have not found their way into the clinical routine of plastic surgery, novel robotic systems customized for microsurgery are aiming to simplify procedures.

## Figures and Tables

**Figure 1 jcm-11-03378-f001:**
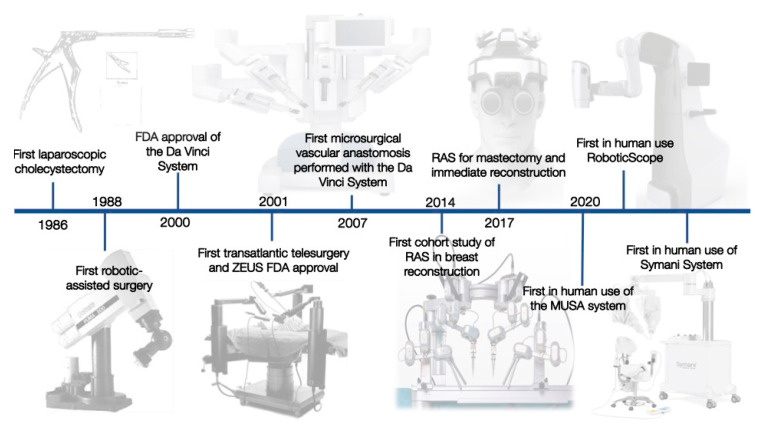
The most important developments in the history of RAS in plastic surgery.

**Figure 2 jcm-11-03378-f002:**
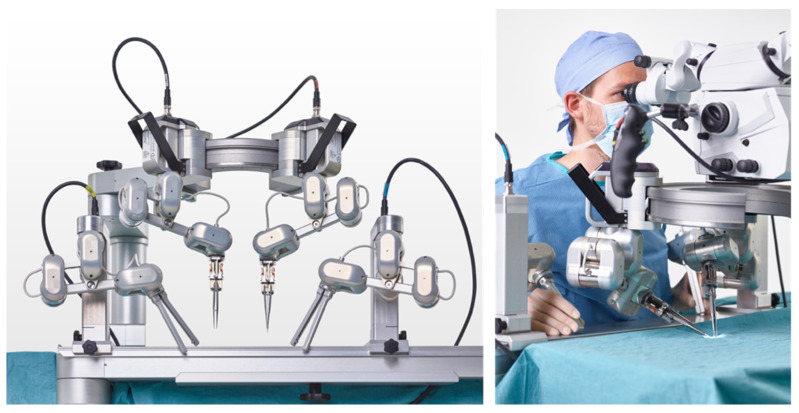
Robotic microsurgery system MUSA from MicroSure (printed with the permission of MicroSure).

**Figure 3 jcm-11-03378-f003:**
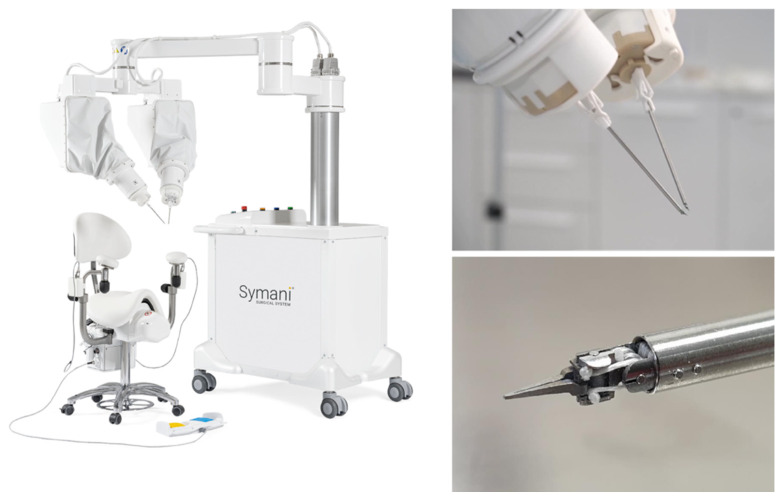
Robotic microsurgery system Symani from MMI (printed with the permission of MMI).

**Figure 4 jcm-11-03378-f004:**
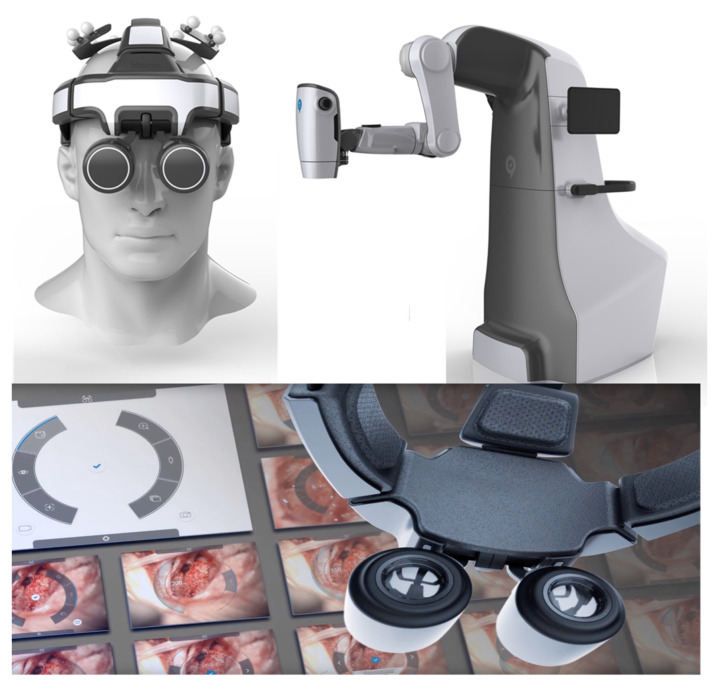
Robotic microscope system RoboticScope from BHS Technologies (printed with the permission of BHS Technologies).

## Data Availability

Not applicable.

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
