# Peer review of "Robotic-Assisted Microsurgery and Its Future in Plastic Surgery"

_jcm, 2022, doi:10.3390/jcm11123378_

Round 1

Reviewer 1 Report

This manuscript reviews the state of robotics development in the field of plastic surgery. The authors have performed a chronological review of the introduction of robotics in surgery, as a prelude to its use in reconstructive surgery. They describe the different robotic systems and explain in greater detail the two robots dedicated specifically to microsurgery and supramicrosurgery: the MUSA (MicroSure) and the Symani (MMI). Authors also review the possibilities of augmented reality, with devices that they have used such as the RoboticScope (BHS Technologies).

Robotic microsurgery is a current and highly important topic in the field of plastic and reconstructive surgery. The topic is relevant,  however, there are some major and minor indications that have to be addressed in the manuscript.

Major indications:

  1. The authors state that it is a systematic review, but there is no reproducible search strategy or protocol described in the manuscript. Furthermore, authors do not assess all results and have not explained the inclusion/exclusion criteria. They do not show a flow diagram describing the articles included or excluded. It looks like a review of the literature but not systematic. Information regarding how the review was carried out is needed. The section “Material and Methods” has to be widely enlarged since there is only written two lines. It is needed to include information on: Protocol and registration, eligibility criteria, information source and search (keywords and Boolean operators), study selection (inclusion or exclusion, flow diagram), data collection process and risk of bias assessment in individual studies. Reproducibility is a major principle of the scientific method.

  1. The Discussion section must explain and evaluate the most relevant findings of the search and results must be compared by citing the original articles. It has to fit with the main research/scientific question of the review. All this should result in reasoning and commenting the different advances supporting the overall conclusion as well. Please, modify the section including references to be able to specifically discuss.

Minor comments:

  1. It is a review of the literature and only one paper from MUSA and one from Symani are cited. Please include existing references for this two robots, as well of references for all the surgical approaches of the DaVinci surgical system, and comment on the findings.

  1. The manuscript does not comment on the transoral robotic reconstruction of oropharyngeal defects. There are several articles on this. Please cite and discuss them.

  1. Since the learning process is commented, it would be advisable to cite studies on learning curves as well as articles or book chapters on Robotic Microsurgical Training to support in a more solid manner the conclusions.

  1. The MMI robot is repeatedly cited in the text by misspelling the name, the correct name of the robotic system is: Symani.

  1. The authors must have the permission of the robot companies to be able to publish the images that are included in this manuscript.

Author Response

This manuscript reviews the state of robotics development in the field of plastic surgery. The authors have performed a chronological review of the introduction of robotics in surgery, as a prelude to its use in reconstructive surgery. They describe the different robotic systems and explain in greater detail the two robots dedicated specifically to microsurgery and supramicrosurgery: the MUSA (MicroSure) and the Symani (MMI). Authors also review the possibilities of augmented reality, with devices that they have used such as the RoboticScope (BHS Technologies).

Robotic microsurgery is a current and highly important topic in the field of plastic and reconstructive surgery. The topic is relevant,  however, there are some major and minor indications that have to be addressed in the manuscript.

We thank the reviewer for acknowledging the importance of our review. 

Major indications:

  1. The authors state that it is a systematic review, but there is no reproducible search strategy or protocol described in the manuscript. Furthermore, authors do not assess all results and have not explained the inclusion/exclusion criteria. They do not show a flow diagram describing the articles included or excluded. It looks like a review of the literature but not systematic. Information regarding how the review was carried out is needed. The section “Material and Methods” has to be widely enlarged since there is only written two lines. It is needed to include information on: Protocol and registration, eligibility criteria, information source and search (keywords and Boolean operators), study selection (inclusion or exclusion, flow diagram), data collection process and risk of bias assessment in individual studies. Reproducibility is a major principle of the scientific method.

We thank the reviewer for this concern. This article represents a review and no systematic review. This was clarified in the article.

  1. The Discussion section must explain and evaluate the most relevant findings of the search and results must be compared by citing the original articles. It has to fit with the main research/scientific question of the review. All this should result in reasoning and commenting the different advances supporting the overall conclusion as well. Please, modify the section including references to be able to specifically discuss.

We thank the reviewer for improving our manuscript. We adapted the Discussion section accordingly.

Minor comments:

  1. It is a review of the literature and only one paper from MUSA and one from Symani are cited. Please include existing references for this two robots, as well of references for all the surgical approaches of the DaVinci surgical system, and comment on the findings.

We thank the reviewer for this insightful comment. A screening of the literature was performed again. References were added- changes are marked in red.

  1. The manuscript does not comment on the transoral robotic reconstruction of oropharyngeal defects. There are several articles on this. Please cite and discuss them.

 We thank the reviewer for improving the quality of our manuscript by highlighting these studies. We implemented them in our manuscript.

  1. Since the learning process is commented, it would be advisable to cite studies on learning curves as well as articles or book chapters on Robotic Microsurgical Training to support in a more solid manner the conclusions.

 We thank the reviewer for this concern. We added a section about the learning curve and cited articles about Robotic Microsurgical training.

  1. The MMI robot is repeatedly cited in the text by misspelling the name, the correct name of the robotic system is: Symani.

 We thank the reviewer for this comment. All changes were made accordingly.

  1. The authors must have the permission of the robot companies to be able to publish the images that are included in this manuscript.

We thank the reviewer for raising this concern. All permissions of the companies are present. To clarify this was added in the figure descriptions.

Reviewer 2 Report

Thank you for the opportunity to review this manuscript. In this study, the authors provide an overview of the emerging technologies, clinical and preclinical studies about, and discusses the potential of robotic assistance in the field of Plastic Surgery. This reviewer believes the following comments need to be addressed:

1- Please discuss the cost of implementing these technologies

2- Discuss strategies to implement these technologies in low, as well as middle or high income countries alike

3- It may be helpful to include a table that summarizes the studies included in this review

Thanks again for the opportunity to review this manuscript.

Author Response

1- Please discuss the cost of implementing these technologies 

We thank the reviewer for this comment and added the following paragraphs to improve the manuscript:

Result section:

  • “The costs for a Symani system are currently in the high six-figure range (as of May 2022), plus around €2,000 for disposable instruments per procedure (reference: Personal communication with MMI)”.

  • “Pricing for the MUSA system will be substantially lower than competitors due to the use of micro-/supermicrosurgery instruments”.

  • “Acquisition costs for the system currently start at around 200,000€ (as of May 2022)(reference: personal communication with BHS).”

Discussion section:

  • “Therefore, well-funded countries will most probably take the lead in advancing robotic microsurgery until improvements in manufacturing and more competition may result in a significant price decrease. If this process is successful, low-income countries may be able to also implement these technologies.“

2- Discuss strategies to implement these technologies in low, as well as middle or high income countries alike 

Again, we thank the reviewer for this comment. The manuscript now read the following paragraph: “In most countries the high costs of such systems, for whose use no remuneration is envisaged for the time being, cannot be cross-financed by funds from regular operations. Costs are therefore one of the greatest obstacles to clinical use, and cost reduction needs to be considered essential for market establishment. In order to still play an active role in the further development of these systems, funding new technologies through the conduct of preclinical and clinical studies is a possibility. The performance of such studies is also essential to demonstrate a possible benefit or advantage as early as possible and to ensure appropriate remuneration for the use of the new technologies in the long term. Therefore, well-funded countries will most probably take the lead in advancing robotic microsurgery until improvements in manufacturing and more competition may result in a significant price decrease. If this process is successful, low-income countries may be able to also implement these technologies.”

3- It may be helpful to include a table that summarizes the studies included in this review 

Thank you for this comment. Following a careful discussion, we believe that the studies included need to be seen in the context of the manuscript and therefore have not included an additional table.

Round 2

Reviewer 2 Report

Tthe authors have successfully addressed the comments raised by the reviewers.

Author Response

We thank the reviewer for improving our manuscript.